# Exact broadband excitation of two-level systems by mapping spins to springs

Jr-Shin Li[1], Justin Ruths[2] & Steffen J. Glaser[3]

Designing accurate and high-fidelity broadband pulses is an essential component in conducting quantum experiments across fields from protein spectroscopy to quantum optics. However, constructing exact and analytic broadband pulses remains unsolved due to the nonlinearity and complexity of the underlying spin system dynamics. Here, we present a nontrivial dynamic connection between nonlinear spin and linear spring systems and show the surprising result that such nonlinear and complex pulse design problems are equivalent to designing controls to steer linear harmonic oscillators under optimal forcing. We derive analytic broadband $\pi/2$ and $\pi$ pulses that perform exact, or asymptotically exact, excitation and inversion over a defined bandwidth, and also with bounded amplitude. This development opens up avenues for pulse sequence design and lays a foundation for understanding the control of two-level systems.

[1] Department of Electrical and Systems Engineering, Washington University in St Louis, St Louis, MO 63130, USA. [2] Departments of Mechanical and Systems Engineering, University of Texas at Dallas, Richardson, TX 75080, USA. [3] Department of Chemistry, Technical University of Munich, 85747 Garching, Germany. Correspondence and requests for materials should be addressed to J.-S.L. (email: jsli@wustl.edu)

**M**anipulating large ensembles of dynamical systems by a single control signal is a common challenging problem in experimental physics, chemistry, biology, and neuroscience; difficult chiefly due to the inherent heterogeneity present in the systems. For example, in spectroscopic applications such as nuclear magnetic resonance (NMR) spectroscopy, optical spectroscopy, magnetic resonance imaging, and quantum computing, experiments are often performed on quantum ensembles on the order of $10^{23}$ rather than individual molecules or atoms[1–7]. These experiments are conducted through a sequence of engineered electromagnetic pulses that are globally broadcast to produce a desired excitation profile or a time evolution of the quantum ensemble. The system Hamiltonian is inherently not uniform over the ensemble due to variations in the values of the parameters that characterize the dynamics of each individual spin. NMR spectroscopy offers canonical examples, including broadband excitation and inversion of spin populations, where only a single coil is available to generate radio-frequency (RF) fields that are used to steer all nuclear spins of a particular type (e.g., protons) in a macroscopic sample from some initial state (e.g., thermal equilibrium) to a desired target state (e.g., transverse magnetization with a desired phase) despite inherent variation in the natural frequency of the spins. The ability to perform a state-to-state transformation of this type has direct impact on experimental outcomes in, for example, increasing medical image resolution[8], amplifying off-resonance signal recovery in large-protein NMR[9], achieving effective coherent control of logical qubits[10], and developing pulses for ultrafast all-optical signal processing devices[11].

The evolution of a two-level system is described by the Bloch model, which can be represented as a bilinear dynamical system. If the spin population is characterized by a single frequency, it is well known that an on-resonance sinusoidal pulse, where the frequency of the RF pulse is tuned to match the frequency ($\omega$) of the sample, is able to excite or invert the spins exactly. In these cases, the RF pulse $\gamma B_{1y}\cos(\omega t)$ achieves equalization (a $\pi/2$ or 90° rotation) or inversion (a $\pi$ or 180° rotation) of the spin population in $T = \pi/(2\gamma B_{1y})$ or $T = \pi/(\gamma B_{1y})$ units of time, respectively[12], where $B_{1y}$ is the strength of the RF pulse applied around the $y$ axis of the rotating frame and $\gamma$ is the gyromagnetic ratio.

In practice, however, the resonance frequencies of the spins are spread over a range due to chemical shifts caused by varying levels of magnetic shielding[1] or by magnetic field gradients[8]. Practitioners regularly use this frequency dispersion in constructive ways in order to distinguish between nuclei in different chemical environments, however, the same phenomenon makes manipulating spin populations uniformly over a specified bandwidth highly nontrivial.

Calculating the time evolution of the spin magnetization corresponding to a given RF pulse can be accomplished through straightforward integration, however, solving the inverse pulse design problem, which seeks to construct an RF pulse that produces a desired distribution of final spin states, or magnetization profile, is much more difficult. Work to date has focused on developing robust numerical optimization techniques to search for an optimal pulse that achieves broadband excitation or inversion. These methods are often highly customized and have slow or unverified convergence rates, especially when designing pulse sequences for difficult experiments with more demanding performance specifications[13]. The development of analytical approaches or optimization-free algorithms for broadband pulse design has been minimal due to the nonlinearity of the spin dynamics. The exceptions are the hyperbolic secant pulse[14], which is a parameter-dependent selective inversion pulse,

where the selectivity is achieved when the amplitude of the pulse reaches above a threshold and when the pulse parameters are appropriately tuned, and the Shinnar–Le Roux algorithm, which maps the problem of selective pulse design to the design of finite impulse response (FIR) filters[15].

In this work, we present an analytic result for broadband pulse design. The main discovery is to reveal a nontrivial dynamic connection between nonlinear spin and linear spring systems under optimal forcing and present the unexpected result that such nonlinear and complex pulse design problems are equivalent to designing controls for steering linear harmonic oscillators, which is analytically tractable. We derive analytic broadband $\pi/2$ and $\pi$ pulses that perform exact, or asymptotically exact, excitation and inversion over a defined bandwidth, and also with bounded amplitude.

## Results

**Mapping spins to springs**. The Bloch model is a semi-classical description of the time evolution of a two-level system and is of the form,

$$\frac{\mathrm{d}}{\mathrm{d}t}\begin{bmatrix} M_x(t,\omega) \\ M_y(t,\omega) \\ M_z(t,\omega) \end{bmatrix} = \begin{bmatrix} 0 & -\omega & u(t) \\ \omega & 0 & -v(t) \\ -u(t) & v(t) & 0 \end{bmatrix}\begin{bmatrix} M_x(t,\omega) \\ M_y(t,\omega) \\ M_z(t,\omega) \end{bmatrix}, \quad (1)$$

when the duration, $T$, of the external RF pulse, $u(t) = -\gamma B_{1y}$ and $v(t) = -\gamma B_{1x}$, is much shorter than the transverse and longitudinal relaxation times $T_2$ and $T_1$[1]. In this case, the effects of relaxation can be neglected and the bulk spin magnetization vector $\mathbf{M} = (M_x, M_y, M_z)$ evolves on a sphere. In the absence of an irradiating RF pulse for a time that is much longer than $T_1$, the spin vector aligns with the static magnetic field, conventionally in the $+z$ direction (the spin magnitude is typically normalized so we consider a unit sphere). To simplify the presentation of the analysis, we consider a pulse applied only along the $y$ axis, i.e., $v(t) = 0$ (we discuss an implementation with both controls in Supplementary Note 5). For the sake of simplicity, the rest of the manuscript uses dimensionless variables, normalized by the maximum RF amplitude $\gamma A$, $|B_{1y}| \le A$ (i.e., RF amplitude and frequency dispersion are measured in units of $\gamma A$, and time is measured in units of $\gamma^{-1}A^{-1}$).

Consider separately the dynamics of an undamped harmonic oscillator represented in matrix form,

$$\frac{\mathrm{d}}{\mathrm{d}t}\begin{bmatrix} x(t,\omega) \\ y(t,\omega) \end{bmatrix} = \begin{bmatrix} 0 & -\omega \\ \omega & 0 \end{bmatrix}\begin{bmatrix} x(t,\omega) \\ y(t,\omega) \end{bmatrix} + \begin{bmatrix} 1 \\ 0 \end{bmatrix}u(t), \quad (2)$$

where the state $\mathbf{X} = (x,y)$, $\omega$ and $u(t)$ represent the oscillator's velocity and position, frequency, and external forcing, respectively (Supplementary Note 1). Observe that the unforced dynamics, $u(t) = v(t) = 0$ in Eq. (1), of the transverse components ($M_x$ and $M_y$) of the spin magnetization coincide with the dynamics of an unforced, $u(t) = 0$, undamped harmonic oscillator, or spring, of the same frequency. It is intriguing then to explore the possibility that the connection between the spin and the spring can be preserved even when driven by a common external input. In this report, we identify and characterize the unexpected dynamic connection between the time evolution of forced spins and springs that is not limited to the linear regime of small rotation angles. We exploit this connection to offer an analytic solution to the broadband pulse design problem. Ultimately, there can be no direct mapping for every point along the evolution of these linear and nonlinear systems; however, we discover a dynamic projection that maps the endpoints of the trajectory of a spin to that of a spring, which is sufficient, because

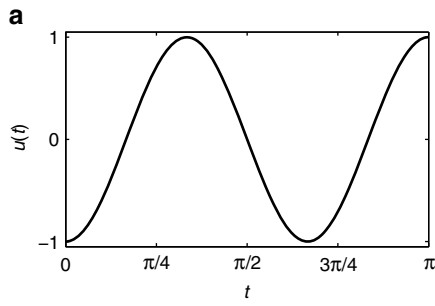
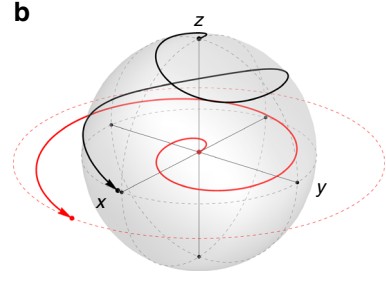

**Fig. 1** Exact excitation of single spins. **a** The minimum-energy control $u^\star_{\pi/2}$ steering the spring from $\mathbf{X}_0 = (0,0)$ to $\mathbf{X}_{\pi/2} = (\pi/2,0)$, with **b** the corresponding trajectories of the spring (*red*) and spin (*black*) for $\omega = 3$ and $T = \pi$

the pulse design problem is defined by the desired terminal magnetization profile.

To develop this connection, we construct the complex projection,

$$f(t) = \frac{M_x(t) + iM_y(t)}{a(t) + M_z(t)}, \qquad (3)$$

where $0 \le t \le T$ is the pulse duration and $a(t) = a_1(t) + ia_2(t)$ is a complex-valued function satisfying a Riccati equation with the initial condition $a(0) = 1$ and depending on the time-varying RF pulse (see Eq. (17) in Supplementary Note 2). If $a(t) = 1$ over the entire duration, then $f(t)$ simply becomes the stereographical projection. Using the fact that the magnitude of the vector $\|\mathbf{M}\| = 1$, we can compose conditions on $f(t)$ and $a(t)$ to ensure that the dynamic projection corresponds to a valid (i.e., noncomplex-valued and unique) Bloch trajectory (Supplementary Note 6). The necessary and sufficient condition for the projection in Eq. (3) to be a one-to-one correspondence to a spin trajectory is in terms of the following bound on $f(t)$,

$$0 \le |f|^2 < \frac{1 - |a|^2 + \sqrt{\left(1 - |a|^2\right)^2 + 4a_2^2}}{2a_2^2}, \qquad (4)$$

where $|a|^2 = a_1^2 + a_2^2$. This condition also indicates why the stereographic projection fails to provide a mapping in the general case (Supplementary Note 6).

Using this dynamic projection, we show that an RF pulse, $u(t)$, which results in $f(t)$ that satisfies Eq. (4) and the following integral condition,

$$\int_0^T u(t)e^{i\omega t}dt = -\frac{\pi}{2}, \qquad (5)$$

steers the Bloch system from $\mathbf{M}(0) = (1,0,0)$ to $\mathbf{M}(T) = (0,0,1)$—a reverse excitation pulse (Supplementary Note 2). Classic linear systems analysis[16] reveals that the external forcing that steers a harmonic oscillator of the same frequency from $\mathbf{X}(0) = (\pi/2,0)$ to $\mathbf{X}(T) = (0,0)$ must satisfy the same integral condition in Eq. (5); therefore, under the bound in Eq. (4), the conditions on the controls for driving the spin and spring coincide. More precisely, we show that $u(t)$ satisfying Eq. (4) will exactly transfer the value of the dynamic projection $f(t)$ from 1 to 0 and hence drives a reverse excitation of the spin in the Bloch system, in the absence of irregular singularities caused by the evolution of $a(t)$ at time $T$ (Supplementary Note 3). The input that steers the spring from $\mathbf{X}_0 = (0,0)$ to $\mathbf{X}_{\pi/2} = (\pi/2,0)$ at time $T$ can be converted to a forward excitation pulse taking the spin from $\mathbf{M}_0 = (0,0,1)$ to $\mathbf{M}_{\pi/2} = (1,0,0)$ by reversing it in time and changing its sign[17].

**Analytical optimal excitation pulses**. Most importantly—and the fundamental result reported in this work—a properly conditioned forcing of the linear harmonic oscillator, which can be computed using known linear systems theory, is an excitation pulse for the nonlinear Bloch system. Among the many potential control functions $u(t)$ that complete the desired transfer, we can select the minimum-energy control, i.e., the control $u^\star(t)$ that minimizes the cost functional $\int_0^T u^2(t)dt$ (Supplementary Note 1). For example, the minimum-energy control that steers the spring with frequency $\omega = 3$ from $\mathbf{X}_0$ to $\mathbf{X}_{\pi/2}$ at $T = \pi$ is given by $u^\star_{\pi/2}(t) = -\cos(3t)$, which is a $\pi/2$ pulse taking the spin from $\mathbf{M}_0$ to $\mathbf{M}_{\pi/2}$. This optimal control and the resulting trajectories of the spring and the spin with $\omega = 3$ are illustrated in Fig. 1.

The same notion can be adopted to design an inversion pulse, which is realized by constructing a control that steers the spring from $\mathbf{X}_0$ to $\mathbf{X}(T) = (\pi,0) = \mathbf{X}_\pi$, or by concatenating a $\pi/2$ pulse with its time-reversed version[17], which is a pulse sequence $\mathbf{X}_0$ to $\mathbf{X}_{\pi/2}$ to $\mathbf{X}_\pi$. The minimum-energy inversion pulse and the resulting trajectories for the spring and the spin of $\omega = 3$ and $T = \pi$ are illustrated in Supplementary Fig. 2.

**Broadband excitation pulses**. The dynamic connection between spin and spring has enabled the analytic design of $\pi/2$ and $\pi$ pulses that manipulate the spin magnetization at a single frequency $\omega$. We now apply this discovery to design a control $u(t)$ that simultaneously steers an ensemble of springs between $\mathbf{X}_0$ and $\mathbf{X}_{\pi/2}$ (or $\mathbf{X}_\pi$), which is called a broadband $\pi/2$ (or $\pi$) pulse, respectively. The minimum-energy broadband controls can be derived by solving the integral Eq. (5) in function space (since $\omega$ becomes a variable) and are composed of the prolate spheroidal wave functions (Supplementary Note 7). Figure 2 and Supplementary Fig. 7 show broadband $\pi/2$ and $\pi$ pulses, respectively, which produce uniform excitation over the designed bandwidth. In practice these pulses can be constructed using the discrete prolate spheroidal sequences available in many scientific programming tools, such as "dpss" in Matlab (Supplementary Note 7).

Practical considerations, e.g., limited power of RF coils, make it critical to design pulses with bounded amplitude. Steering an ensemble of springs with a bounded control is a challenging optimal control problem. However, we show that it can be reduced to a convex optimization problem, which can be effectively solved, and the optimal control has a bang–bang pulse shape (Supplementary Note 7). The bang–bang pulse in Fig. 2 is an example of a bounded amplitude broadband $\pi/2$ pulse. The performance (i.e., average excitation) can be adjusted by selecting different amplitude bounds and pulse durations.

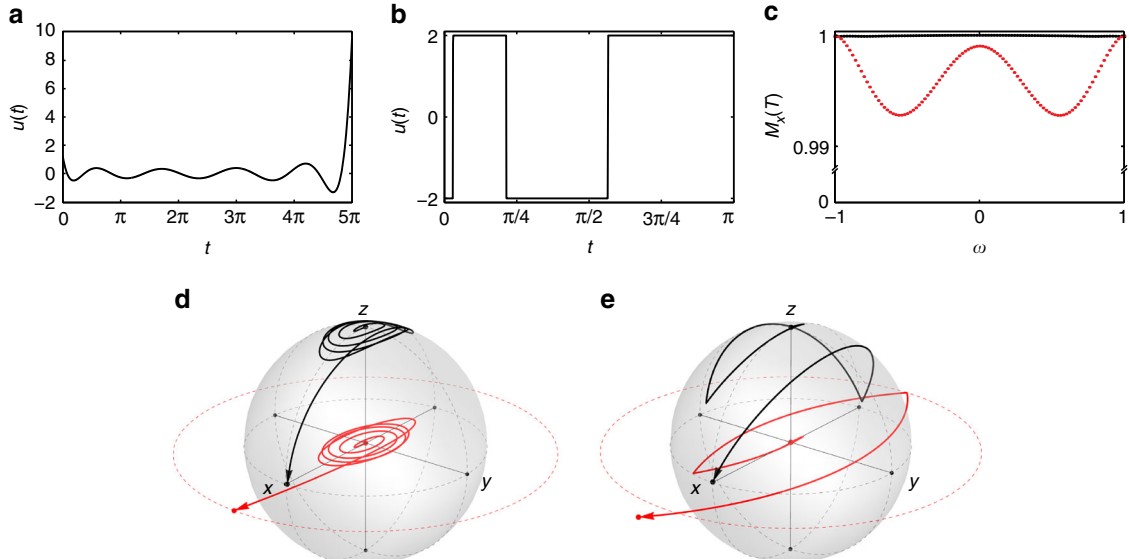

**Fig. 2** Broadband excitation of spin ensembles. The **a** broadband minimum-energy and **b** amplitude-limited controls steering a family of harmonic oscillators with frequencies $-1 \leq \omega \leq 1$ from $\mathbf{X}_0 = (0,0)$ to $\mathbf{X}_{\pi/2} = (\pi/2,0)$ with the corresponding trajectories, **d** and **e**, respectively, of the harmonic oscillator (red) and nuclear spin (black) for $\omega = -1$. These pulses achieve **c** high-fidelity magnetization excitation profiles, $M_x(T)$, over the frequencies $-1 \leq \omega \leq 1$, with average excitation of 1000 (minimum energy in black) and 0.996 (amplitude limited in red), respectively

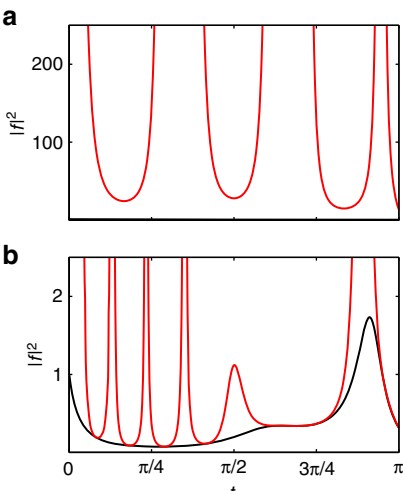

**Fig. 3** Feasibility of pulse designs. The trajectory of $|f|^2$ (black) and the corresponding bound (red) given by the right side of Eq. (4) resulting from the **a** minimum-energy control $u^\star_{\pi/2}$ and **b** a quadratic control, $u(t) = (18t^2 + 4 - 9\pi^2)/8$, that steer the spring from $\mathbf{X}_0 = (0,0)$ to $\mathbf{X}_{\pi/2} = (\pi/2,0)$. In **a** the trajectory of $|f|^2$ is much smaller than the bound, appearing as a *horizontal line* at the *bottom* of the figure

## Discussion

The dynamic mapping in Eq. (3) reveals a nontrivial connection between nonlinear spin and linear spring systems under optimal forcing and enables the design of analytical broadband pulses. The bound on this dynamic projection $f(t)$ in Eq. (4) is critical to ensure the feasibility of the designed pulses. To illustrate the importance of this bound, in Fig. 3 we plot the time evolution of $|f(t)|^2$ for the minimum-energy control and a quadratic control, which satisfies the integral condition in Eq. (5), but not the bound in Eq. (4). Supplementary Fig. 6 shows several other such counterexample controls. Note that all controls steer the spring from $\mathbf{X}_0$ to $\mathbf{X}_{\pi/2}$, but only when Eq. (4) is satisfied in the

minimum-energy case does the control also steer the spin from $\mathbf{M}_0$ to $\mathbf{M}_{\pi/2}$.

Moreover, empirical results presented in Supplementary Note 5 and Supplementary Fig. 5 strongly suggest that the framework described here can be generalized to design pulses that employ two controls simultaneously and also that achieve arbitrary flip angles (not only restricted to $\pi/2$ and $\pi$ pulses).

Analytic control inputs for exact broadband excitation and inversion of spin systems provide a new toolset that will help to push past some of the limitations in using current numerical methods for pulse sequence design. The application of dynamic projection methods to the spin–spring relationship introduced in this article, together with convex optimization methods to solve the amplitude-bounded optimal control problem, lays a foundation to develop pulse sequences for more complicated profiles, such as frequency selective pulses.

## Methods
**Optimal steering of springs.** The minimum-energy control that steers the spring modeled in Eq. (2) from $\mathbf{X}_0$ to $\mathbf{X}_F$ can be derived using least squares theory[16] and is of the form $u^*(t) = \mathbf{B}'e^{-\mathbf{A}'t}\mathbf{W}^{-1}[e^{-\mathbf{A}T}\mathbf{X}_F - \mathbf{X}_0]$, where $\mathbf{W}$ is the controllability Gramian of the spring system, defined by $\mathbf{W} = \int_0^T e^{-\mathbf{A}t}\mathbf{B}\mathbf{B}'e^{-\mathbf{A}'t}dt$, where $\mathbf{A} = \begin{bmatrix} 0 & -\omega \\ \omega & 0 \end{bmatrix}$ and $\mathbf{B} = \begin{bmatrix} 1 \\ 0 \end{bmatrix}$. For example, if the frequency of the spring is $\omega = 3$, then the minimum-energy control that drives the spring from $\mathbf{X}_0 = (\pi/2,0)$ to $\mathbf{X}_F = (0,0)$ of during $\pi$ is $u^*(t) = -\cos(3t)$ for $t \in [0,\pi]$ (Supplementary Note 1).

**Dynamic mapping between spin and spring.** Using the relation $M_x^2 + M_y^2 + M_z^2 = 1$ for all $t \in [0,T]$, the complex projection defined in Eq. (3) follows the dynamic equation

$$\dot{f} = i\omega f + \frac{1}{2}uf^2 + \frac{1}{2}\beta u, \quad f(0) = 1, \qquad (6)$$

where $\beta = e^{2i\omega t}$, if the complex function $a(t)$ is chosen to satisfy

$$\dot{a} = -\frac{u\beta}{2m}a^2 - \frac{uz(\beta - 1)}{m}a + \frac{u(1 + z^2 - z^2\beta)}{2m}, \quad a(0) = 1. \qquad (7)$$

Integrating (6) using contour integration, we show that $f$ is steered to $f(T) = 0$ following the control input that drives the spring from $\mathbf{X}_0 = (\pi/2,0)$ to $\mathbf{X}_F = (0,0)$. This implies that the spin is excited from $\mathbf{M}_0 = (0,0,1)$ to $\mathbf{M}_{\pi/2} = (1,0,0)$, so that this control is a $\pi/2$ (or 90°) pulse (Supplementary Note 2).

**Data availability**. The authors declare that the data supporting the findings of this study are available within the paper and its Supplementary Information.

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

## Acknowledgements

This work was supported by the National Science Foundation under the awards CMMI-1462796 and ECCS-1509342 (JSL), the Air Force Office of Scientific Research under the award FA9550-17-1-0166. (JSL), and the DFG grant GL 203/7-2 of SPP 1601 (SJG).

## Author contributions

J.-S.L. identified the phenomena and constructed main proof; J.R. provided the alternative proof; the authors jointly designed the numerical experiments, and J.-S.L. and J.R. performed the numerical simulations; J.-S.L. and J.R. did primary writing of the manuscript and Supplementary Information, and S.J.G. provided critical comments and intellectual content.

## Additional information

**Competing interests:** The authors declare no competing financial interests.

