## [Peer Review File · Nature Communications]

REVIEWERS' COMMENTS:

Reviewer #1 (Remarks to the Author):

Comments on the paper

Exact broadband excitation of two-level systems: mapping spins to springs

This paper is interesting and must be accepted. It proposes a systematic and partially analytic method to obtain broadband controls steering an initial spin profile towards a target profile in finite time T . This method is based on an original idea: the construction of mappings between the spin trajectories on the Bloch sphere and the trajectories of a classical harmonic oscillator on the plane sharing the small control input. This mapping is non linear and defined by two nonlinear differential equations of first order in the complex plane. Since broadband steering control admits well established solution for linear harmonic oscillator, these solutions can be transposed to spin, up to some mathematical technicalities relative to an algebraic solvability condition and the absence of irregular singularities in one of these two non-linear differential equations defining these mappings.

The fact that, as illustrated on figure 2¹, this method provides with few computations broadband controls $[0, T] \ni u(t)$ steering an ensemble of spins with $\omega \in [-1, 1]$ from $Z = +1$ to $X = +1$ on the Bloch sphere is very interesting from a practical view point. This is to be compared with usual optimal control algorithms such as GRAPE requiring much more computations.

This method seems to be specific to ensemble of spins sharing the same control but with different Larmor frequencies (parameter ω in equation (1)). Can it be extended to a second kind of continuous parameters corresponding to a dispersion of the multiplicative control gain g ((u, v) in equation (1) changed to (gu, gv) with g belonging to an interval around 1) ?

Reviewer #2 (Remarks to the Author):

This paper is concerned with the problem of constructing an optimal electromagnetic pulse to steer an ensemble of quantum particles to a desired quantum state. In the case of two level quantum spin systems, the equations of motion are bilinear differential equations. The main idea of the paper is to map the problem into one involving the control of linear spring equations. This involves the use of a dynamic projection which maps the end points of the spin trajectory with the end points of the spring trajectory. This projection is defined by the solution to a Riccati equation and a necessary and sufficient condition is given for the projected spring trajectory to be in 1-1 correspondence with the spin trajectory. The main significance of this is that it enables an analytic solution to be given for the optimal pulse corresponding to minimizing the control energy.

Overall, the paper provides a very useful new method for constructing optimal pulses for quantum control. Hence, it should be published.

¹I guess that y-label of figure 2a and 2b corresponds in fact to u .

Reviewer #3 (Remarks to the Author):

The work is an important and broadly interesting contribution to control of single two-level systems. It introduces a surprising and useful connection between single axis, unitary control and driven harmonic oscillators. It helps to address the long standing question of control efficiency in two-level systems.

At the same time it is a first step. It does not address the key question of non-holonomic control and leaves open questions on ultimate efficiencies. The paper provides a new isomorphism so that mapping onto the well-studied examples of control of driven harmonic oscillators can lead to new insight, but the controls found via this will not be optimal in a larger setting.

I suggest publication with only very minor changes:

In the abstract please replace "under certain conditions" with a more informative description of the limits.

In the discussion on quantum computing (lines 27,28) please make clear that the averaging is over a dispersion of operators over which there is a limit to the precision of control.

In figure 2 the control is over a spectrum which is not reflected in the figure. A plot of the fidelity (overlap in this case) vs ω would be informative.

Response to Reviewer Comments

Summary

We appreciate the time and effort that the Referees have spent to review our manuscript submission. We are pleased to hear the very positive and constructive comments provided on the originality, novelty, and broad appeal of our work on mapping spins to springs for analytical quantum pulse design. In this letter, we make a point-by-point response to the Referees' questions and requests.

In Response to Referee #1

1. *This method seems to be specific to ensemble of spins sharing the same control but with different Larmor frequencies (parameter ω in equation (1)). Can it be extended to a second kind of continuous parameters corresponding to a dispersion of the multiplicative control gain ε ((u, v) in equation (1) changed to $(\varepsilon u, \varepsilon v)$ with ε belonging to an interval around 1)?*

Compensating for RF inhomogeneity is, indeed, another key obstacle towards high fidelity control of quantum systems. Our framework in its current form cannot be directly applied to design pulses that compensate for RF inhomogeneity, because the integral condition in equation (21) of Supplementary Note 2 will become ε -dependent, where $\varepsilon \in [1 - \delta, 1 + \delta]$, $0 < \delta < 1$, and can not be simultaneously satisfied for all values of ε . However, it is possible that a different dynamic projection could be constructed to map spins to springs to account for RF inhomogeneity. This direction of investigation would be an excellent, although highly non-trivial extension of our work here.

In Response to Referee #3

1. *In the abstract please replace “under certain conditions” with a more informative description of the limits.*

We rearranged the sentences in the abstract so that “under certain conditions” is replaced by “under optimal forcing”.

2. *In the discussion on quantum computing (lines 27,28) please make clear that the averaging is over a dispersion of operators over which there is a limit to the precision of control.*

The following sentence after line 28 was added to make this point more clear:

“The system Hamiltonian is not uniform over the ensemble due to variations in the values of the parameters that characterize the dynamics of each individual system.”

3. *In figure 2 the control is over a spectrum which is not reflected in the figure. A plot of the fidelity (overlap in this case) vs omega would be informative.*

The fidelity vs ω was plotted in Figure 2c.